Methods of body temperature assessment in Conolophus subcristatus, Conolophus pallidus (Galápagos land iguanas), and Amblyrhynchus cristatus X C. subcristatus hybrid

Valle Carlos A. 1
http://orcid.org/0000-0001-8492-5489 Grijalva Colon J. 2
Calle Paul P. 3
Muñoz-Pérez Juan Pablo 2
Quezada Galo 4
Vera Carlos A. 4
Lewbart Gregory A. 5 greg_lewbart@ncsu.edu
1 Universidad San Francisco de Quito USFQ, Colegio de Ciencias Biológicas y Ambientales COCIBA, Campus Cumbayá Av. Diego de Robles S/N e Interoceánica , Quito , Ecuador
2 Galápagos Science Center GSC , Isla San Cristobal, Galápagos , Ecuador
3 Wildlife Conservation Society, Zoological Health Program , New York, NY , USA
4 Technical Biodiversity Research, Dirección Parque Nacional Galápagos , Puerto Ayora, Galápagos , Ecuador
5 North Carolina State University College of Veterinary Medicine , Raleigh, NC , USA
Schwanz Lisa
Electronic publication date: 2019 Feb 4
Publication date: 2019
Volume: 7
Electronic Location ID: e6291
Received 2018 Oct 2; Accepted 2018 Dec 11
Copyright: © 2019 Valle et al.
Copyright year: 2019
Copyright holder: Valle et al.
License: This is an open access article distributed under the terms of the Creative Commons Attribution License, which permits unrestricted use, distribution, reproduction and adaptation in any medium and for any purpose provided that it is properly attributed. For attribution, the original author(s), title, publication source (PeerJ) and either DOI or URL of the article must be cited.
License URL: https://creativecommons.org/licenses/by/4.0/

Keywords: Amblyrhynchus cristatus X C. subcristatus hybrid, Conolophus subcristatus, Conolophus pallidus, Galápagos, Land iguana, Body temperature, Infrared

Funding: The Leona M. and Harry B. Helmsley Charitable Trust This work was supported by The Leona M. and Harry B. Helmsley Charitable Trust. The funders had no role in study design, data collection and analysis, decision to publish, or preparation of the manuscript.

==============================
Since cardiovascular, respiratory, and metabolic systems of reptiles are affected by temperature, accurate measurements are of great importance in both captive husbandry and research. Ectothermic animals generally have core body temperatures close to ambient temperature but can differ from the immediate environment if they are using sunlight to thermoregulate. Many zoological facilities and exotic pet caregivers have begun using infrared temperature guns to assess ambient temperatures of reptile enclosures but there are currently few studies assessing the efficacy of these devices for measuring the body temperatures of reptiles. Conolophus subcristatus, Conolophus pallidus (Galápagos land iguanas), and Amblyrhynchus cristatus X C. subcristatus hybrid are robust land iguanas endemic to the Galápagos archipelago. By comparing the infrared body temperature measurements of land iguanas against virtual simultaneous collection of cloacal temperatures obtained using a thermocouple thermometer, we sought to assess the efficacy of this non-invasive method. We found that internal body temperature can be predicted with a high level of accuracy from three external body temperature sites, providing a good non-invasive method that avoids the capture of animals.

Introduction

Since cardiovascular, respiratory, and metabolic systems of reptiles are affected by temperature, accurate measurements are of great importance in both captive husbandry and research (Sato et al., 1995; Deen & Hutchinson, 2001; Seebacher & Franklin, 2005; Long, 2016). Body temperature assessment is also an important part of veterinary health examinations (Music & Strunk, 2016; Cusack et al., 2018). While ectothermic animals generally have core body temperatures close to ambient temperature, it has been shown that the animal’s temperature can differ from the immediate environment (Raske et al., 2012). The size (mass) of the animal would also effect this as smaller animals heat up and cool down more quickly than larger ones (Gillooly et al., 2001; Seebacher & Franklin, 2005).

Many zoological facilities and exotic pet caregivers have begun using infrared temperature guns to assess ambient temperatures of reptile enclosures (Rizzo, 2015), but there are only a few studies assing the efficacy of these devices for measuring the body temperatures of amphibians and reptiles (Hare, Whitworth & Cree, 2007; Rowley & Alford, 2007; Halliday & Blouin-Demers, 2017). By comparing the infrared body temperature measurements of land iguanas against virtual simultaneous collection of cloacal temperatures obtained using a thermocouple thermometer, we sought to assess the efficacy of this non-invasive method in a large terrestrial reptile.

Conolophus pallidus and C. subcristatus are land iguanas endemic to the Galápagos archipelago. The former species is restricted to Santa Fe Island and both species are classified as vulnerable by the IUCN Red List of Threatened Species (World Conservation Monitoring Centre, 1996a, 1996b). The hybrid has only been documented from South Plazas Island (Rassmann, Trillmich & Tautz, 1997). As part of a population health assessment authorized by the Galápagos National Park (GNP), wild iguanas from three islands (North Seymour, South Plazas, and Santa Fe) were captured in July 2018. Veterinary health examinations, that included sampling blood, ectoparasites, and feces, were performed on each animal in accordance with the ethics and animal handling protocols of Galápagos Science Center (GSC) and the GNP.

Materials and Methods

This study was conducted in the Galápagos archipelago of Ecuador in July, 2018 as part of a population health assessment authorized by the GNP Service (Permit # PC-70-18 to G.A. Lewbart) and approved by the Universidad San Francisco de Quito ethics and animal handling protocol.

A total of 21 adult C. pallidus, 30 C. subcristatus, and one Amblyryhnchus cristatus X C. subcristatus hybrid weighing between 1.2 and 7.2 kg (mean = 3.9 kg) were captured for health assessments. All data were collected over a period of 4 days, with the ambient environmental temperature ranging from 20.5 to 26 °C at the time of examination. Body temperature measurements with a Nubee® infrared temperature gun (Model NUB838OH) were collected from the right axillary space (TAXI), dorsum (TDOR), and right femoral space (TFEM) (Fig. 1). The axillary space is the area just caudal to the insertion of the upper arm to the body. The dorsum is an area directly above the axillary location at the most dorsal part of the animal. The femoral space is the area just cranial to the insertion of the rear limb to the body. The device was held approximately seven cm from each location for 5–10 s (until the temperature stabilized). This device has a distant to spot ratio of 8:1. Thus, the area being measured is approximately one cm. Cloacal temperatures (TINT) were measured using an EBRO® Compact J/K/T/E thermocouple thermometer (Model EW-91219-40; Cole-Parmer, Vernon Hills, IL 60061, USA) inserted approximately 10 cm into the vent. The thermocouple thermometer was held in place until a stable temperature reading could be recorded (approximately 15 s).

Figure 1 Anatomical locations for infrared temperature assessment.

An adult Conolophus subcristatus with the right axillary, dorsum, and right femoral locations for infrared temperature measurement depicted. Photo by GA Lewbart.

Statistical analysis

We used repeated-measures ANOVA with Huynd-Feldt (HF) method to correct for departure from sphericity (HF: ε = 0.7896778, Mauchly’s Tests for sphericity: W = 0.54929, P ≤ 0.001) to test for temperature differences between the four methods. We then performed pairwise comparisons using paired t-test with Bonferroni-Holm correction. Given non-independence of measurements, we investigated the best predictors of internal (a.k.a. cloacal) body temperature (TINT) in two complementary ways. Running simple linear models, we first looked at simple pairwise correlations using as regressors the three infrared external temperatures: Femoral (TFEM), Dorsal (TDOR), and Axillary (TAXI). Then, we chose the external body temperature with the highest correlation coefficient as the best single predictor of internal temperature. We also extracted the first principal component of the three external temperature measures to be used as a single compound predictor. Statistical analyses were performed in R version 3.5.1 (R Development Core Team, 2018).

Results

Average body temperatures measured at each site are displayed in Table 1. Figure 2 graphs all four temperature points for each animal. The average temperature differed significantly between the four methods (repeated-measures ANOVA: F2.4,120.9 = 9.8397, P < 0.001); however, the effect size was small (η2 = 0.01224529). Pairwise comparisons showed significant differences between TDOR and TAXI (t = 3.01, DF = 51, P = 0.012) and all three other pairwise comparisons (P < 0.01 for all cases). The non-significant differences included TINT–TDOR (t = 1.36, DF = 51, P = 0.359) and TDOR–TFEM (t = 0.68, DF = 51, P = 0.498). Simple pairwise correlation between temperature variables showed that all four measures were highly correlated (Fig. 2). Internal body temperature TINT correlated highly and significantly with each of the three methods of recording external body temperature: TFEM (r = 0.967; t = 26.941, DF = 50, P < 0.001), TDOR (r = 0.928; t = 17.63, DF = 50, P < 0.001), TAXI (r = 0.918; t = 16.08, P < 0.001).

Table 1 Average temperature values for the land iguanas in this study.

Anatomic location	Average temperature (°C)	Standard deviation	
Infrared temperature gun TAXI	30.3	3.4	
Infrared temperature gun TDORS	31.1	3.6	
Infrared temperature gun TFEM	31.0	3.4	
Thermocouple thermometer TINT	31.4	3.5	
Note:

Average temperature and standard deviation data for the 52 land iguanas (Conolophus pallidus, C. subcristatus, and Amblyryhnchus cristatus X C. subcristatus hybrid) assessed by cloacal thermocouple thermometer and an infrared temperature gun.

Figure 2 Graphic representation of the land iguana body temperatures.

Body temperature measurements data for the 52 land iguanas (Conolophus pallidus, C. subcristatus, and Amblyryhnchus cristatus X C. subcristatus hybrid) assessed by cloacal thermocouple thermometer and an infrared temperature gun.

The best single predictor (i.e., with the highest correlation coefficient) of internal body temperature (TINT), running a simple linear model was femoral body temperature (TFEM) (r2 = 0.936, F1,50 = 725.8, P < 0.001): TINT=0.754(SE±1.144)+0.989(SE± 0.037)*TFEM

where SE are standard errors of the estimated coefficients.

The predictive model of internal body temperature (TINT), running a simple linear model using the compound variable (Principal component, PC1: r2 = 0.948, F1,50 = 903.8, P < 0.001) from all external body temperatures was (Fig. 3): TINT=31.381 (SE± 0.113)+0.587(SE± 0.020)*PC1

where SE are standard errors of the estimated coefficients.

Figure 3 Bar graph depicting land iguana body temperatures from the different sites.

A simple linear model using the compound variable from all external body temperatures for the 52 land iguanas (Conolophus pallidus, C. subcristatus, and Amblyryhnchus cristatus X C. subcristatus hybrid) assessed by cloacal thermocouple thermometer and an infrared temperature gun.

Discussion

Thermal biology is a rich and dynamic area of biology that studies the way animals react and adapt to their environment by regulating body temperature. Camacho & Rusch (2017) provide a thorough and comprehensive review of the subject and the various methodologies that have been used to determine body temperature in lizards. Dozens of studies exist in reptiles over a range of topics that include preferred body temperature, voluntary thermal selection, and absolute thermal tolerance (Tattersall, 2016; Camacho & Rusch, 2017). Many of these studies use standard thermometers or thermocouples, but some use infrared cameras (Luna & Font, 2013) and others use infrared guns similar to the one used in our study (Hare, Whitworth & Cree, 2007; Rowley & Alford, 2007; Carretero, 2012; Halliday & Blouin-Demers, 2017). Carretero (2012) found that in small lacertid lizards of the genus Podacris, infrared readings were consistently higher than cloacal readings. Our study differs from the previous four works in the important area of animal size. The land iguanas in our study were between 10 times and 100 times larger than the small lizards, amphibians, and snakes in these studies.

In our study the differences between the three methods of recording body temperature were small and all three external body temperatures slightly underestimated internal body temperature. Dorsal temperature was the closest to internal temperature, axillary was the lowest, and femoral was in the middle.

The best model predictor shows that internal body temperature can be predicted with a high level of accuracy from the three external body temperatures, providing a good non-invasive method that avoids the capture of animals. Femoral temperature, in spite of not being the most accurate approximation to internal body temperature, was the best single predictor to estimate internal body temperature as it was the least variable with the highest correlation coefficient. We speculate this is because the femoral area is closest to the thermocouple location (deep cloaca).

One issue to consider, although we did not experience it, would be an animal that has been in a burrow for a period of time and then climbs out into direct sunlight. In this case it’s possible, if not likely, that the external temperature might not reflect the true core temperature. This would be an interesting angle to pursue with either this species or other reptiles. Another factor that should be considered is what is the preferred temperature (Tpref) for the land iguana. Based on the overall health of the animals in this study it may be similar to the Tpref of 25–35 °C for its mainland relative Iguana iguana, the green iguana (Girling & Raiti, 2004).

Conclusions

This non-invasive method for measuring body temperature using an infrared temperature gun is useful for approximating body temperature in Galápagos land iguanas. The quantitative assessment provides data that can be most likely be used for monitoring the health status of other similar sized lizard species in captive collections or in the field. Further evaluation is warranted for its applicability to other species of large reptiles.

Supplemental Information

Supplemental Information 1 Supplementary File 1.

Biological data for the 52 land iguanas including all temperature measurements. Each animal is listed by number and the vertical columns contain all pertinent values.

Click here for additional data file.

We wish to thank the following people and institutions for their support and assistance: This research was authorized by the GNP Service and was conducted with support of The Leona M. and Harry B. Helmsley Charitable Trust, the Galápagos Academic Institute for the Arts and Sciences (GAIAS), Universidad San Francisco de Quito (USFQ), the GSC, the Wildlife Conservation Society, and the Charles Darwin Foundation. We thank Karen Ingerman, Diego Páez-Rosas, Gustavo Jiménez-Uzcátegui, Diego Quiroga, Ken Lohmann, and Kent Passingham for their support. Special thanks to the GSC staff: Carlos Mena, Stephen Walsh, Philip Page, Sofia Tacle, Sylvia Sotamba, Karla Vasco, and Soledad Sarzosa. DPNG staff: Edison Muñoz, Daniel Lara-Solis, Jorge Carrión-Tacuri, Ingrid Jaramillo, and Maryuri Yépez. In addition, we thank the Galápagos National Park (GNP) for the request and trust granted for sampling, and Galápagos Science Center (GSC) for the logistic support during the study. Exceptional thanks to the captain Yuri Revelo and the deckhand Estalin Guamanquishpe “Bacalao.”

Additional Information and Declarations

Competing Interests

Author Contributions

Animal Ethics

Field Study Permissions

Data Availability

The authors declare that they have no competing interests.

Carlos Valle performed the experiments, analyzed the data, contributed reagents/materials/analysis tools, prepared figures and/or tables, authored or reviewed drafts of the paper, approved the final draft.

Colon J. Grijalva conceived and designed the experiments, performed the experiments, analyzed the data, approved the final draft.

Paul P. Calle performed the experiments, analyzed the data, contributed reagents/materials/analysis tools, prepared figures and/or tables, authored or reviewed drafts of the paper, approved the final draft.

Juan Pablo Muñoz-Pérez performed the experiments, approved the final draft.

Galo Quezada performed the experiments, approved the final draft, facilitated and supervised the field work.

Carlos A. Vera performed the experiments, approved the final draft, facilitated and supervised the field work.

Gregory A. Lewbart conceived and designed the experiments, performed the experiments, analyzed the data, contributed reagents/materials/analysis tools, prepared figures and/or tables, authored or reviewed drafts of the paper, approved the final draft.

The following information was supplied relating to ethical approvals (i.e., approving body and any reference numbers):

This study was approved by the Universidad San Francisco de Quito ethics and animal handling protocol.

The following information was supplied relating to field study approvals (i.e., approving body and any reference numbers):

This study was conducted in the Galápagos archipelago of Ecuador as part of a population health assessment authorized by the Galápagos National Park Service (Permit # PC-70-18 to G.A. Lewbart).

The following information was supplied regarding data availability:

Raw data are available in a Supplemental File.

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
