# Peer review of "Methods of body temperature assessment in Conolophus subcristatus, Conolophus pallidus (Galápagos land iguanas), and Amblyrhynchus cristatus X C. subcristatus hybrid"

_PeerJ, doi:10.7717/peerj.6291_

## Round 0.1 · original submission · Major Revisions

This paper presents a methodological validation of measuring body temperature, motivated by health and husbandry applications. Both reviewers agree that the data collection is valid and the data are robust. However, they raise large concerns with the presentation of the study that need to be addressed before the manuscript can be considered further. They also provide many stylistic suggestions that would improve the paper.

If you decide to revise, particularly address:

1. Reviewer #1's advice for alternative statistics. The external measurements are not independent and should not be entered as independent predictor variables in a multiple regression.
2. Pairwise scatterplots would be a much-better way to visualize the data with respect to the scientific question. Currently, Fig 2 is not an appropriate way to present the data and Fig 3 is not very useful.
3. Although this paper is designed to be short and targeted, I agree that it needs more contextual information in the introduction and discussion. Both should include additional text and citation of ectotherm thermoregulation, particularly with respect to behavioural thermoregulation and thermal inertia. The discussion should have implications for how the method might work in other contexts - for the iguana and for other species.

·

Basic reporting

The writing is clear, as are figures. Methods are adequately described, and the data are provided.

Experimental design

The sampling design also appears fine. I have some concerns with the analyses used (see below).

Validity of the findings

The findings seem valid (with the caveat of the statistical issues) and uncontroversial. Some more thought needs to go into the context in which the findings might apply beyond this system, however.

Additional comments

Nice neat manuscript.

63: it might be worth mentioning that the degree of variance from the environment temperature is affected by the size of the animal (its thermal inertia) as well as how quickly the environmental temperature shifts. Interesting points here, because you have a large reptile (high inertia, so potentially will deviate to a large extent), but it is in a fairly stable thermal environment (i.e., equatorial island).

85: since these devices average temperature over an area, it might be worth working out what size surface area is measured when you hold your particular IR gun at 5-7cm from the animal.

92: "departure from sphericity". What is that, exactly? If you are talking about a univariate response variable (i.e. temperature), then departure from normality would be the more usual criterion.

95: Unusual also to report a standard linear model (i.e. normal error distribution) as a glm. Not incorrect, just unusual. If you are reporting it as a glm, then pls specify error distribution (presumably, normal) and link function (presumably identity). Alternative analysis pathway (below) would obviate this point.

Using multiple regression with highly correlated predictor variables is problematic, and can lead to massively inflated standard errors on coefficients (and so parameter estimates that jump all over the place). I think rather than a multiple regression (or glm) you should probably just use simple pairwise correlation coefficients (r, plus plots) to show that all four measures are highly correlated. You could then do both of a) choose the external temperature with the highest r-value as your best single predictor of internal temperature, and b) extract the first principal component of external temperature measures as a compound variable. You then run a simple linear model to predict internal temperature from a) the best single predictor, and b) the compound external temperature measure.

I don't think you need p-values anywhere in this manuscript. Except, maybe, to show the correlations are all significantly different from 0. Much more important are the parameter estimates (r, and partial coefficients) and their SEs.

In the Discussion, I think you need to put some thought into how reliable the internal/external comparison would be if animals were subject to a rapid temperature change (e.g., in/out burrows; in/out of shade). This goes to the generality of your findings -- under what conditions would you expect internal and external temperatures to align?

Reviewer 2 ·

Basic reporting

This paper is very poorly written, and does not meet the most basic standards required for a scientific paper.

There is no significant background or context to place the study within the scientific literature. The authors should examine papers comparing other thermal methods in a similar manner to see how to better situate their study and its results.

The figures do not provide enough information to allow us to evaluate the science, they are presented in a naive way.

The paper does not address any thermal literature at all (even the methodological literature) which is substantial. The authors should read this literature to see how to present their data.

Experimental design

1. This study could probably provide interesting information to the scientific community.
2. Moderately well defined, but not really situated, stated how it fills a gap for 2 species of lizard, but could be broader.
3. Data collection seems to be fine, but presentation is inadequate.
4. Methods okay

Validity of the findings

1. This is apparently not replication of exisiting results.

2. Data is probably robust and sound.

3. Conclusions are not situated in exisiting research, and difficult to evaluate because research presentation is so poor.

4. Speculation is not relevant here.

Additional comments

Review of Methods of body temperature assessment in Conolophus subcristatus and Conolophus pallidus (Galápagos land iguanas).
Title - Broaden to include largish land reptiles more generally?

Abstract - does the MS have any broader significance? I would not start with the study animal, but instead with the second sentence, eventually introducing the land iguanas as a study system.
Line 30 - I do not see 'captivity' and 'research' as mutually exclusive.
Lines 30-32 - I think the majority of research on ectotherms emphasizes that their body temperatures can be quite different from ambient because they behaviourlly thermoregulate - this sentence seems misleading
Lines 32-35 - emphasise that they are large - we have done this on (much) smaller reptiles and it did not work as a method, because too much of the background was measured by the 'gun'

Introduction

Again, do not start with the study animal
Line 60 - do not contrast research & captivity - what do you really mean? For care and data collection on wild animals? Something like that?

Methods

Study dates?
Line 91 - are repeated measures the best way to determine this? It seems to me that you have compared 4 different methods of taking the temperature on one iguana - maybe better to control for variation due to individual using a random factor in the model? That would save you from Mauchley's test which is overly conservative when sample sizes are small…

Line 93 - between 2 things, among many

Line 93-94 if you have a significant ANOVA there is no need to Bonferroni correct post-hoc tests

Results

In general, in the figure and table captions, it is not necessary to put 'bar graph of' or 'line graph of' - it is obvious what kind of graph it is. Start with the data "Means ± SDs of xxxxx"

Table 1: It would be good to report more than just the standard deviations here - ranges are probably the most useful - more importantly, I think Fig 3 reports the same data. It would be better to show a box-and-whisker plot of each of these so we can get a feel for the data distribution of each measurement in relation to the others, and get rid of table 1 all together

Figure 2 - what is the x-axis of figure 2? Iguana number? It is difficult to see what information we can derive from this graph. It is hard to see the overlapping lines. If you plotted the box and whisker plot recommended for figure 1, that would help us see the bias of different methods, if there were any. Another graph might be to show all 4 dots for each iguana (no line - there is no reason to connect these), which might show variation among methods within animals, which might be interesting.

Discussion

Line 123-126 - the slight underestimation of the 3 external Tbs, and their relative positions should be obvious from the figures, but I don't see it, the figures have to be targeted to help the audience see this

Lines 127 - 131 - why was femoral temperature better than dorsal temperature as a predictor? Was it less variable? More explanation is needed here

There needs to be more discussion of outcomes of the different methods - did one method work well on cold days? What about hot days? Why were some individuals super cold and others very hot in your data?

What is iguana preferred temperature? Were they able to reach it? How is this relevant to their health status if this is not reported? How different were they from ambient?

---

## Round 0.2 · Minor Revisions

In my assessment, the revisions to the manuscript address the reviewer comments. The new statistical analyses and high-quality Figs 2 and 3 have greatly improved the presentation of the results.

I have a few, small editorial revisions prior to acceptance:

Line 54: remove ‘like land iguanas’.
Move species paragraph to be the last paragraph of the introduction.
Lines 117-120: I would suggest keeping the text on the pairwise comparisons that ARE significant, and then saying the other two comparisons were not significant (p> 0.05)
Move the paragraph lines 165-170 to follow the paragraph ending line 156
Line 161 & 163, Change to ‘preferred temperature’ and ‘Tpref’. ‘Preferred optimal temperature zone’ is not a term I have seen in the thermal literature before (notwithstanding it begs the question as to whether the preferred temperature is the optimal temperature)
Figure 3 legend needs to be amended.

---

## Round 0.3 · Minor Revisions

In reviewing your resubmission (version 2), I noticed that the final manuscript file has not been replaced with the revised version - it is still v1. Please upload the newest revision and resubmit.

---

## Round 0.4 · accepted · Accept

Thank you for addressing the reviewers comments. As an expert in the field of reptile thermoregulation, I am satisfied that the revisions have adequately addressed the reviewers' concerns regarding statistical analyses, context and presentation. The manuscript is now suitable for publication.

#